# Liquid Biopsy Detecting Circulating Tumor Cells in Patients with Non-Small Cell Lung Cancer: Preliminary Results of a Pilot Study

**DOI:** 10.3390/biomedicines11010153

**Published:** 2023-01-07

**Authors:** Maria Giovanna Mastromarino, Sara Parini, Danila Azzolina, Sara Habib, Marzia Luigia De Marni, Chiara Luise, Silvia Restelli, Guido Baietto, Elena Trisolini, Fabio Massera, Esther Papalia, Giulia Bora, Roberta Carbone, Caterina Casadio, Renzo Boldorini, Ottavio Rena

**Affiliations:** 1Division of Thoracic Surgery, Ospedale Maggiore della Carità di Novara, 28100 Novara, Italy; 2Department of Medical Sciences, University of Ferrara, 44121 Ferrara, Italy; 3Tethis S.p.A., 20143 Milan, Italy; 4Division of Pathology, Ospedale Maggiore della Carità di Novara, 28100 Novara, Italy; 5Department of Health Sciences, University of Piemonte Orientale, 28100 Novara, Italy

**Keywords:** liquid biopsy, non-small cell lung cancer, circulating tumor cells, early-stage lung cancer, next generation sequencing, cancer screening

## Abstract

Lung cancer is still the leading cause of cancer-related death worldwide. Interest is growing towards early detection and advances in liquid biopsy to isolate circulating tumor cells (CTCs). This pilot study aimed to detect epithelial CTCs in the peripheral blood of early-stage non-small cell lung cancer (NSCLC) patients. We used Smart BioSurface^®^ (SBS) slide, a nanoparticle-coated slide able to immobilize viable nucleated cellular fraction without pre-selection and preserve cell integrity. Forty patients undergoing lung resection for NSCLC were included; they were divided into two groups according to CTC value, with a cut-off of three CTCs/mL. All patients were positive for CTCs. The mean CTC value was 4.7(± 5.8 S.D.) per ml/blood. In one patient, next generation sequencing (NGS) analysis of CTCs revealed v-raf murine sarcoma viral oncogene homolog B(BRAF) V600E mutation, which has also been identified in tissue biopsy. CTCs count affected neither overall survival (OS, *p* = 0.74) nor progression-free survival (*p* = 0.829). Multivariable analysis confirmed age (*p* = 0.020) and pNodal-stage (*p* = 0.028) as negative predictors of OS. Preliminary results of this pilot study suggest the capability of this method in detecting CTCs in all early-stage NSCLC patients. NGS on single cell, identified as CTC by immunofluorescence staining, is a powerful tool for investigating the molecular landscape of cancer, with the aim of personalized therapies.

## 1. Introduction

Lung cancer is still the leading cause of cancer mortality worldwide [1]. It is often diagnosed at an advanced stage due to inadequate screening methods, the late onset of symptoms and the delayed referral of patients to examinations, with a 5-year OS rate of only 20% [2]. Focusing on early detection may be the best chance to cure this malignancy. Currently, lung cancer screening is performed using a high-resolution computed tomography (HRCT) set at a low dose. This screening is recommended for people at high-risk (heavy smokers, age >50 years, or patients with chronic obstructive pulmonary disease (COPD)) [3,4]. Screening high-risk persons with LDCT can reduce lung cancer mortality but also causes false-positive results, leading to unnecessary tests and invasive procedures, overdiagnosis, incidental findings, and an increase in distress.

Interest is growing in biological tests that may improve cancer screening. Among these, advances are being made in liquid biopsy methods to isolate circulating CTCs. CTCs generally refer to cancer cells in the peripheral blood that have disseminated from the primary tumor or metastatic sites. Several studies have shown that CTCs can be detected in the blood of patients with early-stage lung cancer, particularly in stage I disease [5]. CTCs are responsible for the hematogenous cancer spread and the formation of distant metastases. However, their rarity, estimated at one CTC per billion blood cells, presents the biggest technical barrier to their functional studies and clinical application [6]. Recent progresses in CTC isolation technology have allowed for the reliable capture of CTCs from the blood samples of cancer patients. The ability to derive clinically relevant information from isolated CTCs could allow the early diagnosis of tumors.

This pilot study aimed at detecting epithelial CTCs in the peripheral blood of early-stage NSCLC patients. We used the SBS^®^ slide (Tethis, Milan, Italy), a nanoparticle-coated slide, which provides the rapid immobilization of all vital nucleated cells as a homogeneous monolayer, avoiding both pre-selection and maintaining cell integrity and biological properties, allowing immunostaining-based detection of epithelial CTCs [7]. The second endpoint focused on the potential correlations between CTCs count with patients’ clinical characteristics and the pathology of the tumor.

## 2. Materials and Methods

### 2.1. Patient Cohort and Sample Collection

Between November 2017 and December 2018, a total of 40 patients undergoing surgical lung resection for NSCLC at our institution were enrolled in this pilot study. Patients of both genders, older than 18 years of age, with confirmed or suspected NSCLC, who underwent resection were eligible to enter the study. Patients with synchronous malignancy were excluded. Pathological report confirmed the nature of the lesion and NSCLC staging was performed according to the eighth edition of the tumor, node, metastasis (TNM) classification [8]. Demographic, clinical, and pathological data were prospectively collected. A 7.5 mL blood sample was obtained from each patient prior to surgery, with the tumor still in place and collected in K2-EDTA vacuum tubes (Becton, Dickinson and Company, Franklin Lakes, NJ, USA). To reduce the risk of contamination, the first 2 mL of blood drawn from the peripheral vein were not collected. The mean time between sample collection and analysis was 48 h.

This study was approved by the AOU Maggiore della Carità di Novara Institutional Review Board (approval number, CE 105/18). All procedures were performed according to Helsinki Declaration principles. Each patient signed a written informed consent before data collection.

### 2.2. Red Blood Cells (RBCs) Lysis

Each blood sample (7.5 mL) was divided into three 15 mL Falcon tubes (2.5 mL blood per tube) in which 7.5 mL/tube of 1x RBCs lysis buffer were added. The tubes were incubated at 4 °C for 5 min and centrifuged at 1200 rpm for 5 min to collect the nucleated cells. After the pellet washing, the samples were resuspended in 1 mL of phosphate-buffered saline (DPBS1X, Euroclone, Italy). The concentration of nucleated cells was determined using a Burker counting chamber. The Trypan Blue was adopted to evaluate cells’ vitality.

Following the RBCs lysis, the sample was allowed to adhere on SBS^®^ slides (10 to 15 per blood sample), then fixed in 4% paraformaldehyde and permeabilized.

### 2.3. SBS^®^ Slide Technology

We have developed the SBS^®^ slide, characterized by a biocompatible surface, by depositing, under high vacuum, a supersonic seeded beam of titanium oxide (TiOx) clusters produced by a pulsed microplasma cluster source (PMCS). Physical analysis shows that these films possess, at the nanoscale, a granularity and porosity mimicking those of typical extracellular matrix structures and adsorption properties that could allow surface functionalization with different macromolecules, such as DNA, proteins, and peptides [9,10].

SBS^®^ slide has the size of a standard microscope slide, it is transparent, and has no autofluorescence background. The adhesion area of the slide has a unique property of being able to quickly immobilize, in a spontaneous way and with high efficiency, all types of living cells without any special treatment for either slide or cells. The permanent adhesion of viable cells, which takes 20 min thanks to the formation of integrin-based focal adhesion points, is performed at room temperature (R.T.) in a shear-free fashion and does not impair their antigenicity nor their ability to function. The adhesion efficiency on SBS^®^ slide and the sample integrity has been described previously [7].

The performance of SBS^®^ slide has been validated by spiking experiments. To prove that cells with adherent properties, such as epithelial ones and, therefore, also potential CTCs, can also be efficiently immobilized on SBS slide, we performed spiking experiments with Hela S3 cells, a cancer cell line adapted to grow both in suspension and in adherence, which was used as source of mock CTCs. The spiking test was performed by adding Hela S3 cells at different concentrations to a HD blood. To that end, 0, 5, 20, 40, and 300 Hela S3 cells were spiked into 2.5 × 106 of white blood cells (WBC) and seeded in triplicate in 2.5 × 106 WBC/slide concentration on SBS slides. After fixation, the slides were stained with antibodies targeting CD45 and pan-Cytokeratin (PK), counterstained with 4′-6-diamidino-2-phenylindole (DAPI), and scanned with the automated scanning platform.

The assay linearity was evaluated by plotting the average number of Hela S3 cells detected for each of the spike-in concentrations from three replicate slides against the calculated Hela S3 spike-in concentrations, and the linear regression was calculated.

As is shown in Figure 1, the assay was revealed to be linear across all the spike-in concentrations tested (R^2^ = 0.9992). To evaluate the reproducibility of Hela S3 spiking and recovery in blood samples, we tested 2 different spike-in concentrations (5 and 500 cells spiked into 2.5 × 106 of WBC) in 8 replicates. Data showed good reproducibility of mock CTC recovery on SBS slides, with a CV of 27.3% and 11.6% for 5 and 500 cells, respectively.

### 2.4. Immunostaining Protocol

To define a cell as CTC, we set the presence of the epithelial feature (PK positivity) and another specific tumoral biomarker, thyroid transcription factor-1 (TTF-1), as essential requisites.

The epithelial marker is used to identify those rare epithelial cells that circulate in the blood. The presence of epithelial cells in blood can be caused by conditions other than oncological conditions, such as stress, recent diagnostic procedures, or inflammation. Consequently, the application of tumor biomarkers becomes mandatory for the correct identification of CTCs.

TTF-1 is a master regulatory transcription factor for tissue-specific genes and is normally expressed in the thyroid, lung, and brain during embryogenesis [11]. TTF-1 marker is used in immunohistochemistry (IHC) to define whether a biopsy tissue has neoplastic characteristics. TTF-1 expression by IHC in primary lung cancers is dependent on the histological subtypes, as demonstrated by 75–90% of adenocarcinomas (ADCs), 40% of large cell carcinomas, and 0–38% of squamous cell carcinomas. TTF-1 is commonly used to distinguish primary lung ADCs from tumors of other origins that have metastasized to the lung.

In order to align the new SBS technology with this standard method, we chose to use the TTF-1 marker to stain, in immunofluorescence, the patients in the study.

The immunostaining protocol consisted in a primary screening of putative CTC epithelial cells selected for the absences of CD45 and positivity of PK. For each sample, 5 slides were prepared, and the CTCs were identified by fluorescence immunostaining on nucleated cells. Slides were prepared as follows: after resting for 1 h with 5% Blocking Buffer Normal Goat Serum (NGS, Cell Signaling, USA-5425S), the slides were incubated with the primary antibodies, anti-mouse PK (Cell Signaling, USA-4545S) to detect epithelial cells, anti-mouse CD45-647 (BioLegend, USA-393406) for hematopoietic cells, and anti-rabbit TTF-1 (Abcam, UK-ab76013-100) in a subset of patients to detect the tissue origin of the targets.

Then, the slides were washed with DPBS1X and subsequently incubated with secondary antibodies, anti-mouse IgG-Alexa Fluor^®^ 488 (Cell Signaling, USA-4408S) and anti-rabbit IgG-Alexa Fluor^®^ 594 (Cell Signaling, USA-8889S). After washing with DPBS1X, the nuclei were stained with DAPI at R.T. Next, slides were washed with DPBS1X and finally with distilled water. After drying at room temperature, the slides were mounted with glycerol-based mounting medium with DABCO anti-fade (Merck Group, Italy-D27802-100G).

All antibodies used for immunofluorescence were previously tested to determine their optimal specificity and working conditions (such as dilution, incubation time, and temperature) according to the staining of model cells. Examples of anti-CD45 and anti-PK staining on SBS^®^ slide are shown in Figure 2.

To evaluate the specificity of the TTF-1 assay, anti-TTF-1 antibody was tested on high H1975 (lung ADC), low A549, and H1703 (lung ADC and lung squamous cell carcinoma, respectively) TTF-1 expressing cell lines and negative control MCF7 (breast cancer) cell line, as shown in Figure 3. Nuclear-localized staining was observed in expressing cells stained with anti-TTF-1, whereas no staining was observed in negative control cell. Threshold for positivity in NSCLC patients has been set according to the staining of the hematopoietic cells, negative for TTF-1 (TTF-1 positive if mean intensity was >0.20) [12].

### 2.5. CTCs Identification by Automated Microscopy Platform

Epithelial CTCs (defined as PK-positive and CD45-negative cells) were detected with an automated fluorescence microscope. The MetaSystems^©^ Neon_Metafer platform (MetaSystems, Heidelberg, Germany), combined with a Zeiss^©^ fluorescence microscope, allowed to acquire slides in continuous mode and managed sensitive data by generating a final report for each case created. The system generated a gallery of images ascribable to each cell, identified according to the defined acquisition parameters. CTCs were later isolated by laser capture microdissection (Apotome 2 Axio Observer Z1) and characterized for gene mutations by NGS analysis (Oncomine™ Comprehensive Assay v3, ThermoFischer), after WGA amplification (Ampli1 WGA, Menarini, Italy). One patient was analyzed by Sanger sequencing and digital polymerase chain reaction (PCR) (Qiagen, USA).

### 2.6. Statistical Analysis

Patients were divided into 2 groups according to CTC value per ml of blood, with a cut-off of 3 CTCs/mL [13,14]. The data were reported comparing them for groups, such as mean with standard deviation (S.D.), median (I and III quartiles) with range for continuous variables, and percentages (absolute numbers) for qualitative variables. CTC detection levels were analyzed to identify any association with demographic and clinico-pathological characteristics of NSCLC patients. The Wilcoxon–Kruskal–Wallis test was performed for continuous variables and Pearson’s chi-square test for categorical variables. Kaplan–Meier curves and Cox Regression analyses were used to identify predictors of OS and progression-free survival (PFS). Statistical significance was fixed at *p*-values less than 0.05. Statistical analysis was performed using the R 3.6 system [15].

## 3. Results

### 3.1. Patient Characteristics

A total of 40 patients were enrolled into this single-center, prospective, pilot study. The median age was 71 years (range 42–82), and 75% (n = 30) were males. Ninety percent (n = 36) of patients had a history of smoking habit, 22% (n = 9) were current smokers, and 68% (n = 27) former smokers. A professional exposure characterized 40% (n = 16) of the patients. The main histology was adenocarcinoma (68%, n = 27), followed by squamous cell carcinoma (20%, n = 8) and others (12%, n = 5). Among adenocarcinoma histologic subtypes, the majority was represented by the intermediate grade, acinar, or papillary (70%, n = 19). More than half of the patients (65%, n = 26) had a tumor size ≤ 30 mm and in a peripheral location (60%, n = 24). Forty-eight percent of patients (n = 19) had a pathological (p) Stage I and 32% (n = 13) a p-Stage II. A p-nodal (pN) upstaging was observed in 25% (n = 10) of patients (p-N1). None of the patients had distant metastasis at the time of blood sample collection and surgery. The demographic and clinico-pathological characteristics of the study cohort are shown in Table 1.

### 3.2. CTC Detection

Epithelial CTCs were detectable in all 40 NSCLC cancer patients (100%). Forty-five percent (n = 18) of patients had >3 detectable CTCs. CTCs count ranged from 0.2 to 22.2 per mL of blood, with a mean CTC value of 4.7 (± 5.8 S.D.) per mL/blood. WBCs were also enumerated, with a median value of 36.1 × 10^6^/mL of blood (range 15.0–169.0).

### 3.3. CTCs and TTF-1 Immunostaining

TTF-1 expression analysis was performed on 55% (n = 22) of liquid biopsy samples. Representative images of different biomarker profiles expressed by CTCs detected in NSCLC patients’ blood are shown in Figure 4.

Ninety-five percent (n = 21) of tested patients expressed both PK and TTF-1, with a mean PK+/TTF-1+ value per mL/blood of 5.4 (range 0.4–22.2). Specifically, all 13 patients tested with CTC count > 3/mL blood were TTF-1 positive on liquid biopsy, although there was no statistically significant difference with patients with a CTC count < 3/mL (*p* = 0.35). The comparison with cancer tissue biopsy revealed TTF-1 positive concordance in all those patients who underwent IHC on surgical specimens.

Only six patients had CTCs that were exclusively PK-/TTF1+, with a trend toward more advanced tumors (66.7%, n = 4, pT2-T3 versus 33.3%, n = 2, pT1, *p* = 0.713) (Figure 5).

### 3.4. Correlation between CTCs Count and Clinical and Pathological Characteristics

Patients with CTCs >3/mL blood were younger than the group with CTCs <3/mL (median age 68.5 years, range 42–78, versus 71.5 years, range 63–82, *p* = 0.047). There was no relationship between CTCs count and other patient demographics, including gender (*p* = 0.067), smoking history (*p* = 0.073), professional exposure (*p* = 0.897), COPD expressed as forced expiratory volume (FEV1) percentage (*p* = 0.685), hypertension (*p* = 0.822), cardiovascular disease (*p* = 0.842), diabetes (*p* = 0.122), new second malignancy (*p* = 0.433). Data are summarized in Table 2.

Pooled analysis of tumor characteristics revealed no significant correlation between the number of CTCs and tumor size (*p* = 0.842), cancer location (*p* = 0.243), p-stage (*p* = 0.613), pTumor (pT)-stage (*p* = 0.916), nodal status (*p* = 0.886). A trend toward a major rate of squamous cell carcinoma was observed in patients with CTCs >3/mL blood compared to the other group (33%, n = 6, versus 9%, n = 2, *p* = 0.057). No significant differences were found with regard to adenocarcinomas; likewise, a subgroup analysis of this histotype was performed but it was not able to detect any differences regarding intermediate- and high-grade patterns (*p* = 0.974 and *p* = 0.59, respectively). Conversely, low-grade adenocarcinoma showed a correlation with CTCs count >3/mL blood (*p* = 0.05).

### 3.5. Gene Expression and BRAF Mutation

NGS analysis on microdissected CTCs was performed in half the population of the study (50%, n = 20), and it revealed gene variants associated with tumors in all these patients (100%), independently from the number of CTCs detected (*p* = 0.525). Eleven out twenty (55%) patients presented mutations in genes specifically related with NSCLC. One patient carried BRAF V600E mutation, which has also been identified in the tissue biopsy (Figure 6). Sanger sequencing on CTCs revealed mutation at the level of T1799A, leading to the substitution of valine for glutamic acid (V600E). The tissue biopsy of the same patient has been analyzed by NGS panel: 2 out of 31 reads of the BRAF gene showed the mutated allele at position chr7.140453136 A > T (V600E). Finally, both the tissue biopsy and CTCs have been analyzed by digital PCR for the BRAF gene, showing the mutated allele at position chr7.140453136 A > T (V600E).

### 3.6. CTCs Count Associated with Survival

The study cohort was followed for a mean of 20 ± 8 months. Patients with less than three CTCs had a longer OS compared to patients with three or more CTCs, even if a statistically significant difference was not observed (median OS 29.1; 95% CI, 25.5–32.5, versus 27.8; 95% CI, 23.1–32.4 months, *p* = 0.74). Similarly, the group with CTCs >3/mL showed a worse PFS compared to the group with CTCs <3/mL, although a statistical significance was not reached (median PFS 27.4; 95% CI, 22.6–32.3, versus 29.1; 95% CI, 25.6–32.4 months, *p* = 0.829). During the follow-up period, 10% (n = 4) of patients had a local recurrence while 18% (n = 7) presented distant metastases; 15% (n = 6) died of the disease. Nevertheless, the number of CTCs detected did not show any influence on local recurrence (*p* = 0.429) nor distant metastases (*p* = 0.376). Two years OS was 76% in patients with CTCs > 3/mL compared to 77% in CTCs < 3/mL group, *p* = 0.753. One-year PFS was better in patients with CTCs < 3/mL compared to the group with three or more CTCs (94% versus 83%, respectively), while 2 years PFS showed a reversal trend (75% versus 78%, respectively, *p* = 0.43) (Figure 7).

### 3.7. Cox Regression Analysis for Predictors of Survival

Independent univariate analysis showed that age and pN-stage significantly affected OS (*p* = 0.041 and *p* = 0.023, respectively). Multivariable analysis confirmed both these variables as negative predictors of survival (*p* = 0.020 and *p* = 0.028, respectively). These results indicated that increasing age and pN-stage caused a 1.4- and 5.1-fold risk of death, respectively. Regarding PFS, independent univariate analysis showed that pT-stage and p-Stage were significant prognostic factors (*p* = 0.013 and *p* = 0.037, respectively), but multivariable analysis did not confirm these results (*p* = 0.151 and *p* = 0.081, respectively). The investigation of secondary study endpoints revealed no influence of CTCs count, CTCs positivity for TTF-1, nor gene mutations on survival outcomes. Univariate and multivariable analysis results are shown in Table 3.

## 4. Discussion

Lung cancer is the leading cause of cancer-related mortality worldwide for both men and women; therefore, an early detection is paramount to attempt to cure the disease [1]. Screening for NSCLC has become a major public health concern. In addition to standard HRCT, a growing interest in biological tests (so-called “liquid biopsy”) to detect biomarkers of early cancer development has recently emerged. Several studies have examined different blood components, including CTCs, which have been found to be related to cancers and to carry genetic abnormalities similar to the primary tumor [6]. However, only in the last few years have reliable and effective methods in detecting CTCs been developed [5].

Traditional blood filtration method based on the size of the CTCs followed by morphological characterization using classic immunocytochemistry criteria of malignancy (Isolation by Size of Epithelial Tumor Cell –ISET^®^- Rarecells Diagnostics, Paris, France) [16,17] has been progressively replaced by CellSearch^®^ technology (Menarini Silicon Biosystems, Bologna, Italy) [13]. The latter is an immunoaffinity-based assay that isolates CTCs by positively selecting an epithelial cell surface marker, the epithelial cell adhesion marker (EpCAM) [18]. EpCAM is a membrane glycoprotein highly expressed in the majority of carcinomas; accordingly, this is the first and only clinically validated U.S. Food and Drug Administration (FDA)-cleared blood test for detecting CTCs in patients with metastatic breast, colorectal, or prostate cancer [19]. Nevertheless, Chemi and colleagues have recently demonstrated the possibility of detecting CTCs from the pulmonary veins of patients with early-stage NSCLC [14].

In this pilot study, we used a non-enrichment method for CTCs detection from the peripheral blood of patients with NSCLC, which exploits the feature of Tethis proprietary slide, SBS^®^ slide, which provides rapid immobilization at room temperature of all vital nucleated cells as homogeneous monolayer. This slide allows to maintain cell integrity and biological properties, providing immunostaining-based detection and molecular single analysis [7]. Being a label-independent technique, it provides the possibility to efficiently capture CTCs, which have switched to a mesenchymal phenotype [20]. We managed to detect CTCs in all 40 early-stage NSCLC patients, confirming the high sensitivity of the technology for identifying and isolating epithelial cells circulating in blood. Forty-five percent of study cohort had three or more detectable CTCs. The prognostic value of CTCs count has been evaluated in patients with early- and late-stage NSCLC [5,13]. However, no consensus exists regarding the cut-off value associated with the number of CTCs predictive of a negative prognosis, which ranges from 1 to 50 CTCs in the different studies, according to the detection method used and the tumor stage [13,14,21,22].

To better characterize the origin of CTCs and correlate their detection with NSCLC, we performed TTF-1 immunostaining of liquid biopsy samples. TTF-1 is expressed exclusively in thyroid epithelial cells and in bronchioloalveolar cells of the lung; therefore, its expression is used to identify thyroid or pulmonary cancer. Studies using IHC on formalin-fixed paraffin-embedded NSCLC tissues reported that TTF-1 is expressed in 75–90% of adenocarcinomas, 40% of large cell carcinomas, and 0–38% of squamous cell carcinomas [11]. The negative prognostic role of TTF-1 mRNA-positive CTCs detected in the peripheral blood of surgically resected NSCLC patients has been recently demonstrated [23]. Despite the limitations of a partial analysis, 95% of patients expressed both PK and TTF-1, supporting the validity of the cut-off value established for detecting CTCs of NSCLC origin.

Interestingly, the analysis revealed that six patients had CTCs that were exclusively PK-/TTF-1+, with a trend toward more advanced tumors. The ability to detect TTF-1 expression of both PK-/CD45- and PK+/CD45- circulating cells on the SBS^®^ slide allows for increased prognostic power in identifying TTF-1+ patient samples, irrespective of cytokeratin expression, and could demonstrate epithelial plasticity. As a matter of fact, the biomarker profile PK-/TTF-1+ is suggestive of epithelial–mesenchymal transition (EMT), the biological process in which epithelial cells lose their characteristics and acquire mesenchymal features [24,25]. EMT and its intermediate states have recently been identified as crucial drivers of organ fibrosis and tumor progression, conferring upon these epithelial cells properties critical to invasion and distant dissemination, including remarkably increased motility, invasiveness, and the ability to degrade components of the extracellular matrix components [26]. According to this evidence, in our cohort, the PK-/TTF-1+ profile was associated with more advanced tumors (pT2-T3). The inclusion of PK-negative CTCs substantially increased the sensitivity of TTF-1+ CTC detection across all groups. A very recent publication by Jou et al. [27] has shown a new automatic platform based on microfluidic devices able to identify and isolate TTF-1+ CTCs and then successfully perform single-cell sequencing.

Regarding the association between CTCs count with demographic and clinico-pathological characteristics, patients with three or more detected CTCs were younger than the group with lower levels of CTCs (*p* = 0.047), and this finding might be related to a more aggressive disease in younger people. No relationship has been found between other patient demographics and CTCs levels; in particular, the tumor size (*p* = 0.842), the central or peripheral cancer location (*p* = 0.243), and the p-stage (*p* = 0.613) did not correlate with a higher number of detected CTCs in liquid biopsy. On the other hand, a higher CTCs count proved to be effective in identifying early-stage NSCLC, as demonstrated by the correlation between CTCs count >3/mL blood with squamous cell carcinoma (*p* = 0.057) and low-grade adenocarcinoma (*p* = 0.05), in line with the evidence in the literature [22,28].

As expected in cancer patients, age and pathological nodal status were negative predictors of OS, with a 1.4- (*p* = 0.020) and 5.1-fold (*p* = 0.028) risk of death related to increasing age and pN-stage, respectively.

CTCs genotyping may be helpful for a personalized precision medicine. Identifying genetic variations in individuals can be used to predict disease risk, with the potential to halt or retard disease progression. In the last decade, the ability of plasma NGS to detect a wide range of genomic alterations has been demonstrated [28]. In more than half of the patients in which NGS analysis was performed on microdissected epithelial CTCs, we found mutations in genes specifically related with NSCLC. Interestingly, one patient carried BRAF V600E mutation, which occurs in only 1–4% of lung adenocarcinoma [29]. The presence of this mutation has been found to be associated with increased responsiveness to combined therapy with oral inhibitors of BRAF and MEK [30]. Moreover, the BRAF V600E mutation can occur as a resistance mutation for epidermal growth factor receptor—tyrosine kinase inhibitor (EGFR-TKI) therapy. We also identified the same gene mutation detected in liquid biopsy in the cancer tissue specimen of the patient. This concordance is another notable point in favor of the high sensitivity of the methodology adopted in our work to detect CTCs derived from NSCLC. Furthermore, it could pave the way to future scenarios in which the use of liquid biopsy and molecular single cell analysis could help in the rapid identification of BRAF mutational status to drive target anti-BRAF therapy.

Several studies have also established the burden of CTCs as an independent prognostic factor in NSCLC [31,32]. In particular, high CTCs count before, during, or after surgery seems to correlate with a worse prognosis [28]. Our results are suggestive of a correlation with both OS and PFS, although we did not reach statistical significance (*p* = 0.753 and *p* = 0.43, respectively), possibly due to the underpowered feature of the study related to the small sample size.

Recent studies have underlined the relevance of periodic monitoring of CTCs status (before surgery, after treatment, at relapse) to better characterize tumor evolution and the consequent correlation with therapy and clinical parameters [21,28,33]; in addition, the possibility of this technology being able to identify cells with different biomarkers could overcome the limitation of simple CTC enumeration.

This study has some limitations. First of all, the small sample size, without a control group of healthy donors. Second, it is a single-center series with a relatively short follow-up period and a single evaluation of CTCs before surgery. Moreover, TTF-1 immunostaining and molecular analysis was not performed in the entire cohort. Nevertheless, within these limits, our work presents the important strength of being a prospective preliminary study, demonstrating the efficacy of SBS^®^ technology in detecting epithelial CTCs in early-stage NSCLC patients, including CTCs, which have switched to a mesenchymal phenotype. Hence, there is a clear need for further validation of the association between CTCs count and clinico-pathological characteristics in a larger patient cohort. Moreover, the identified CTCs need to be characterized with larger panels of specific biomarkers (such as p63, p40) to confirm their tumor origin, differentiating NSCLC-derived CTCs from other circulating epithelial cells.

## 5. Conclusions

In contrast to a tissue specimen that offers a snapshot of the tumor at a given time and location, liquid biopsy has the potential to overcome both spatial and temporal cancer heterogeneity and can explore the genetic makeup of the patient. Despite CTCs detection being challenging, due to their rarity and difficulty in isolation, we successfully identified circulating epithelial cells in all study cohort. These findings support the current evidence that CTCs can also be isolated in early-stage NSCLC patients. The choice of CTCs detection technology to be used should always consider epithelial plasticity. NGS or other molecular technology on single cells may be useful for investigating the genetic landscape of cancer, providing the possibility of personalized therapies. Our results obtained using the SBS^®^ slide may pave the way for the opportunity to perform more and larger multi-center studies in order to define valid and robust biomarkers, allowing for the optimization of NSCLC early diagnosis. CTCs detection should be improved as a method of “pre-screening” to help the selection of patients who may benefit from HRCT, thereby avoiding unnecessary radiological examinations.

Since the starting of the study, this method has been improved by the development of an instrument (See.d, Tethis S.p.A) that provides complete automation and standardization of the sample preparation on SBS^®^ slides (i.e red blood cell lysis, sample seeding, and fixing), allowing the handling of blood a few hours after collection, thus improving sample informativity and the quality of the pre-analytical step. The deployment of this instrument at different clinical sites will enable the standardization of sample preparation on SBS^®^ slides in multi-center trials.

The next step will be to start a multi-center study to include the analysis and identification of CTCs in a healthy population and cases at risk in order to assess an efficient screening method and hopefully be able to detect NSCLC early.

## Figures and Tables

**Figure 1 biomedicines-11-00153-f001:**
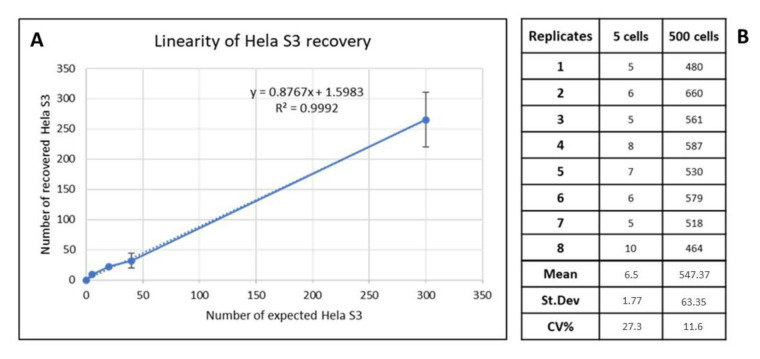
(**A**) Dashed and solid lines represent the assay linearity: plot of recovered Hela S3/slide (*y*-axis) versus the theorical number of Hela S3/slide (*x*-axis). The linear regression was calculated. (**B**) Assay reproducibility calculated for 5 Hela S3 and 500 Hela S3 spiked per slide. St. Dev, standard deviation; CV, coefficient of variation.

**Figure 2 biomedicines-11-00153-f002:**
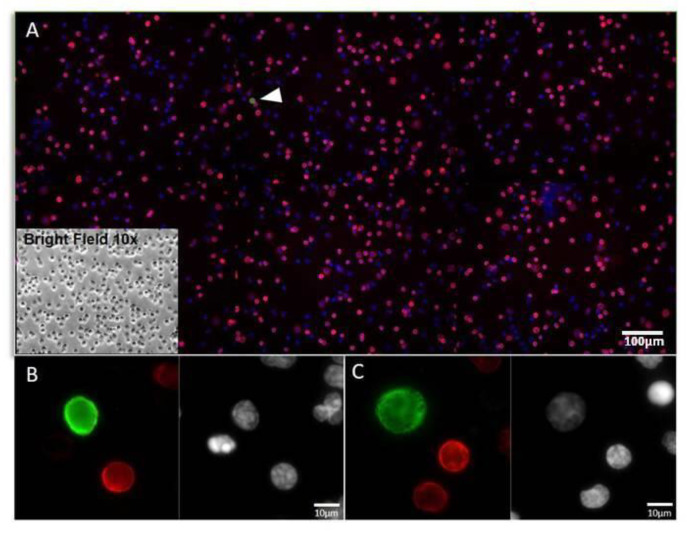
Example of a liquid biopsy sample staining with CD45 and PK in NSCLC patient. (**A**) Representative image at 10× magnification of WBCs and CTCs (white arrow) on SBS^®^ slide stained by immunofluorescence: CD45 (Red); PK (Green); Nuclei (Blue). In the box, brightfield 10× image of the same cells. (**B**,**C**) Examples of CTCs cells acquired at 40× magnification. WBCs, white blood cells; CTCs, circulating tumour cells; PK, pan-Cytokeratin.

**Figure 3 biomedicines-11-00153-f003:**
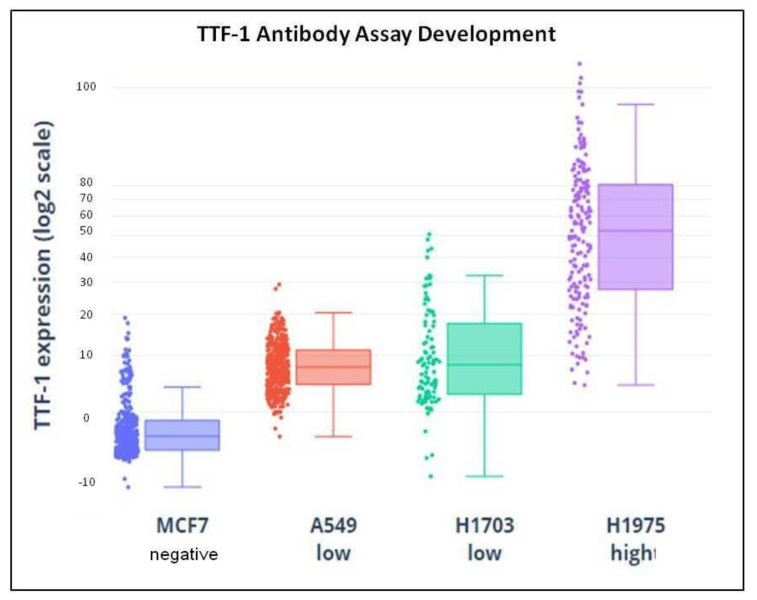
Demonstration of TTF-1 assay specificity: TTF-1-specific antibody was tested in negative MCF7 (breast cancer), low A549 and H1703, (lung ADC and lung squamous cell carcinoma, respectively) and high H1975 (lung ADC) TTF-1-expressing cell lines. Individual cellular TTF-1 immunofluorescence signal is quantified and plotted. No specific staining was seen in negative control cell lines. TTF-1, thyroid transcription factor-1; ADC, adenocarcinoma.

**Figure 4 biomedicines-11-00153-f004:**
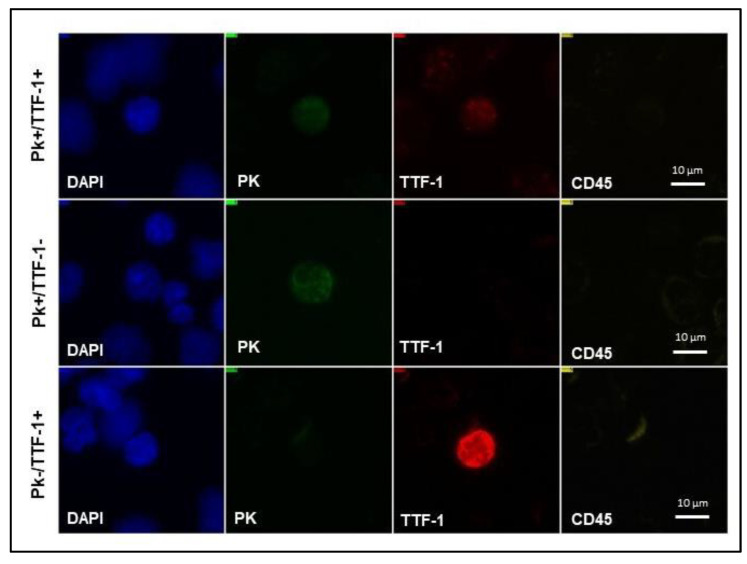
NSCLC CTC subpopulations detected on SBS-CTCs slide. Representative images of CTCs with various biomarker profiles detected on SBS^®^ slide in NSCLC patients’ blood. NSCLC, non-small cell lung cancer; CTCs, circulating tumor cells; SBS^®^ slide, Smart BioSurface^®^ slide; PK, pan-cytokeratin; TTF-1, thyroid transcription factor-1; DAPI, 4′-6-diamidino-2-phenylindole.

**Figure 5 biomedicines-11-00153-f005:**
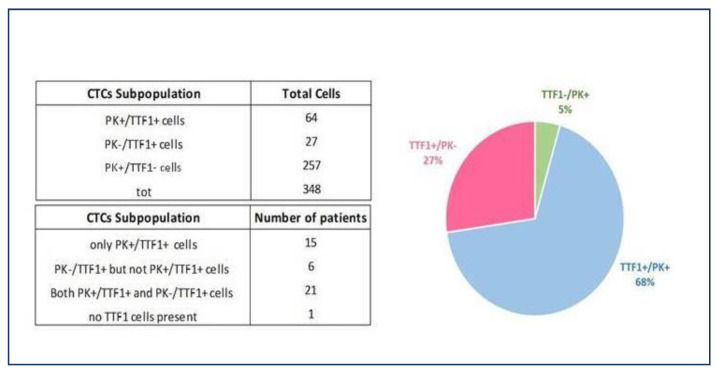
NSCLC patients tested for TTF-1 expression on liquid biopsy. The ability to detect TTF-1 expression of both PK-/CD45- and PK+/CD45-circulating cells on the SBS^®^ slide allows for increased prognostic power in identifying TTF-1+ patient samples irrespective of cytokeratin expression and demonstrates epithelial plasticity. NSCLC, non-small cell lung cancer; TTF-1, thyroid transcription factor-1; PK, pan-cytokeratin; CTCs, circulating tumor cells.

**Figure 6 biomedicines-11-00153-f006:**
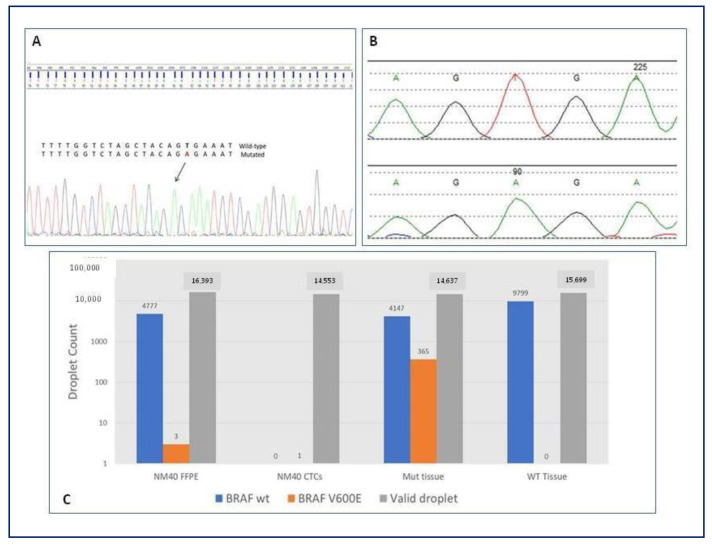
BRAF status on CTCs and tissue biopsy of adenocarcinoma patient. (**A**) Sanger sequencing on microdissected CTCs revealing mutation at the level of T1799A, leading to the substitution of valine for glutamic acid (V600E). (**B**) NGS analysis on tissue biopsy revealing 2 out of 31 reads of the BRAF gene showing the mutated allele at position chr7.140453136 A > T (V600E). (**C**) Digital PCR performed for both the tissue biopsy and CTCs showing the mutated allele at position chr7.140453136 A > T (V600E). BRAF, v-raf murine sarcoma viral oncogene homolog B; CTCs, circulating tumor cells; NM40, patient identification number; GTG > GAG, substitution of valine for glutamic acid; Mut, mutated; WT, wild-type; FFPE, Formalin-Fixed Paraffin-Embedded tissue; NGS, next generation sequencing; PCR, polymerase chain reaction.

**Figure 7 biomedicines-11-00153-f007:**
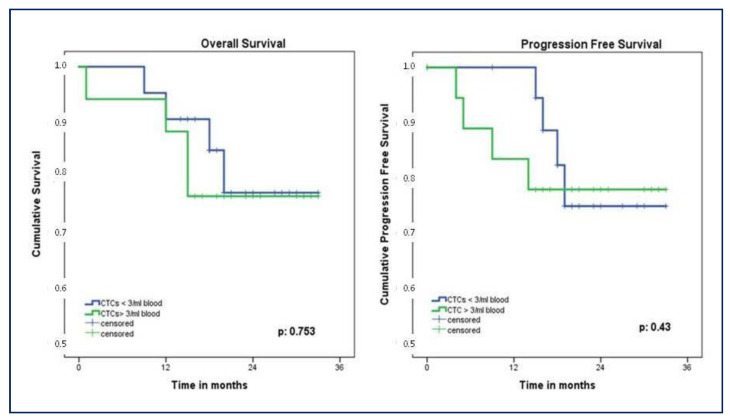
Kaplan–Meier showing OS and PFS according to CTCs count. OS, overall survival; PFS, progression-free survival; CTCs, circulating tumor cells.

**Table 1 biomedicines-11-00153-t001:** Demographics and clinico-pathological characteristics of study cohort.

Characteristics	NSCLC * Patients (n = 40)
Age, years (median, range)	71 years (range 42–82)
Sex, n (%)	
Male	30 (75%)
Female	10 (25%)
Smoking history, n (%)	
No	4 (10%)
Former	27 (68%)
Current	9 (22%)
Professional exposure, n (%)	
No	24 (60%)
Yes	16 (40%)
Hypertension, n (%)	17 (42%)
Cardiovascular disease, n (%)	14 (35%)
Diabetes, n (%)	7 (18%)
Tumor size, n (%)	
<30 mm	26 (65%)
>30 mm	14 (35%)
Tumor location, n (%)	
Central	16 (40%)
Peripheral	24 (60%)
Pathology, n (%)	
Adenocarcinoma	27 (68%)
Squamous cell carcinoma	8 (20%)
Others	5 (12%)
Adenocarcinoma patterns, n (%)	
Low-grade (lepidic)	2 (7%)
Intermediate-grade (acinar or papillary)	19 (70%)
High-grade (solid, micropapillary, or complex gland)	6 (23%)
p-Stage, n (%)	
Stage I	19 (48%)
Stage II	21 (52%)
p-T Stage, n (%)	
T1	20 (50%)
T2	16 (40%)
T3	4 (10%)
p-N Stage, n (%)	
N0	30 (75%)
N1	10 (25%)
CTC ^†^/mL blood	
median, range	1.8 (0.2–22.2)
mean ± S.D. ^§^	4.7 ± 5.8

* NSCLC, non-small cell lung cancer; S.D. ^§^, standard deviation; ^†^ CTC, circulating tumor cell.

**Table 2 biomedicines-11-00153-t002:** Demographics and clinico-pathological patients characteristics according to CTCs count.

Characteristics	CTCs * Count (mL/blood)		*p*-Value
	<3	>3	
Age, years (median, range)	71.5 years (range 63–82)	68.5 years (range 42–78)	0.047
Sex, n (%)			
Male	19 (86%)	11 (61%)	0.067
Female	3(14%)	7 (39%)	
Smoking history, n (%)			
Never	3 (14%)	1 (6%)	0.073
Former	17 (77%)	10 (56%)	
Current	2 (9%)	7 (39%)	
Professional exposure, n (%)			
No	13 (59%)	11 (61%)	0.897
Yes	9 (41%)	7 (39%)	
COPD ^#^, FEV1 (%) ^§^			
< 30%	1 (4.5%)	0 (0%)	0.685
30–60%	1 (4.5%)	3 (17%)	
60–80%	6 (27%)	4 (22%)	
>80%	14 (64%)	11 (61%)	
Hypertension, n (%)	9 (41%)	8 (44%)	0.822
Cardiovascular disease, n (%)	8 (36%)	6 (33%)	0.842
Diabetes, n (%)	2 (9%)	5 (28%)	0.122
Tumor size, n (%)			
≤30 mm	14 (64%)	12 (67%)	0.842
>30 mm	8 (36%)	6 (33%)	
Tumor location, n (%)			
Central	7 (32%)	9 (50%)	0.243
Peripheral	15 (68%)	9 (50%)	
Pathology, n (%)			
Adenocarcinoma	17 (77%)	10 (56%)	0.145
Squamous cell carcinoma	2 (9%)	6 (33%)	0.057
Others	3 (14%)	2 (11%)	0.81
Adenocarcinoma patterns, n (%)			
Low-grade (lepidic)	0 (0%)	2 (20%)	0.05
Intermediate-grade (acinar or papillary)	12 (71%)	7 (70%)	0.974
High-grade (solid, micropapillary or complex gland)	5 (29%)	2 (20%)	0.59
p-Stage, n (%)			
Stage I	9 (41%)	10 (56%)	0.613
Stage II	13 (59%)	8 (44%)	
p-T Stage, n (%)			
T1	10 (45%)	10 (55%)	0.916
T2	9 (41%)	7 (39%)	
T3	3 (14%)	1 (6%)	
p-N Stage, n (%)			
N0	16 (73%)	14 (78%)	0.886
N1	6 (27%)	4 (22%)	
New second malignancy, n (%)			
Yes	1 (4.5%)	2 (11%)	0.433
Patients with CTCs TTF-1 ° +, n (%)			
(Analysis performed on 22/40 patients)	8 (89%)	13 (100%)	0.35

* CTCs, circulating tumor cells; ^#^ COPD, chronic obstructive pulmonary disease; ^§^ FEV1, forced expiratory volume in 1 s; ° TTF-1, thyroid transcription factor-1.

**Table 3 biomedicines-11-00153-t003:** Prognostic factors for overall survival and progression-free survival.

	Univariate Analysis		Multivariable Analysis	
Variable and OS */PFS ^#^	HR ° (95%CI ^§^)	*p*-Value	HR ° (95%CI ^§^)	*p*-Value
**Variable and OS ***				
**Age**	**1.172 (1.007–1.364)**	**0.041**	**1.377 (1.053–1.800)**	**0.020**
**Gender**	34.023 (NA ^†^)	0.294		
**pT-stage**	1.132 (0.459–2.789)	0.788		
**pN-stage**	**0.326 (0.125–0.854)**	**0.023**	**5.047 (1.194–21.341)**	**0.028**
**p-Stage**	2.177 (0.883–5.370)	0.091		
**Pathology**	1.884 (0.544–6.528)	0.318		
**Adenocarcinoma Grading**	4.039 (0.725–22.504)	0.111		
**CTCs ^a^ count** (absolute number)	0.911(0.715–1.161)	0.452		
**CTCs ^a^ count** (>3 vs. < 3/mL blood)	1.247 (0.311–4.990)	0.755		
**CTCs ^a^ TTF1+ ^b^**	40.082 (NA ^†^)	0.275		
**Variable and PFS ^#^**				
**Age**	1.176 (0.941–1.470)	0.154		
**Gender**	36.380 (NA ^†^)	0.433		
**pT-stage**	**3.313 (1.291–8.502)**	**0.013**		0.151
**pN-stage**	15.235 (NA ^†^)	0.198		
**p-Stage**	**8.122 (1.129–58.429)**	**0.037**		0.081
**Pathology**	1.414 (0.246–8.123)	0.698		
**Adenocarcinoma Grading**	4.043 (0.599–27.309)	0.152		
**CTCs ^a^ count** (absolute number)	0.804 (0.514–1.257)	0.339		
**CTCs ^a^ count** (>3 vs. <3/mL blood)	0.404 (0.042–3.894)	0.433		

* OS, overall survival; ^#^ PFS, progression-free survival; ° HR, hazard ratio; ^§^ CI, confidence interval; ^†^ NA, not applicable; ^a^ CTCs, circulating tumor cells; ^b^ TTF-1, thyroid transcription factor-1.

## Data Availability

The data presented in this study are available on request from the corresponding author.

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
