# Peer review of "Liquid Biopsy Detecting Circulating Tumor Cells in Patients with Non-Small Cell Lung Cancer: Preliminary Results of a Pilot Study"

_biomedicines, 2023, doi:10.3390/biomedicines11010153_

Round 1
Reviewer 1 Report
The authors show preliminary results on detecting CTCs in NSCLC patients using a commercial SBS slide.
There are 4 main comments to improve the work:
1. The performance of SBS slides should be validated (by spiking cancer cells in blood) and presented in the article.
2. A control group of healthy candidates should be added to the results.
3. The reason why only PK and TTF-1 biomarkers were chosen in this research should be explained in more detail.
4. Why authors used anti-mouse antibodies for human CTCs detection? Cat. # for each antibody must be provided.
Author Response
Dear Reviewer,
Thank you very much for your time, valuable and stimulating comments.
- We have improved our work with a new paragraph in Materials and Methods, in which we have described the SBS Slide technology and its validation (2.3 SBS ® Slide Technology text lines 98-138)
-
As explained within the limits of the study, unfortunately we do not have a control group of healthy donor candidates (text lines 447-448). We are planning more and larger studies, including healthy volunteers.
- We have improved the paragraph 2.4 Immunostaining protocol explaining the choice of PK and TTF-1 biomarkers in text lines 139-158.
-
The antibodies used in common immunological techniques are produced from different animal species, the most used antibodies are from mouse and rabbit. The technical data sheet of each antibody specified which species it reacts with, and examples of staining are shown. According to datasheet of PK antibody, used in this study, is present reactivity with human, mouse, rat and monkey species. To prove this statement, it was tested in immunofluorescence on the human cell line Hela. Moreover, we have provided Cat.# for each antibody (text lines 163-181).
The revisions to our manuscript have been marked up using the “Track
Changes” function of MS Word.
We have added a new figure, figure 1, so all other figures have been updated up to number 7.

Reviewer 2 Report
This study described a cohort of 40 lung cancer patients whose CTCs were tested by a customized platform the authors devised, using immunoaffinity to detect those expressed with pan-cytokeratin and TTF-1 followed by downstream NGS single cell sequencing. This is a very interesting work and deserves publication. The strong point of this ms is that they did include the analyses regarding CTC and the prognosis. However, the weak point is they actually did not described the so called SBS slide. The main hurdle of CTC liquid biopsy is the capture efficiency as well as the reproduciblity and reliablity. This platform looks promising in this small cohort and I do think the readers will love to know in more detail what is this SBS slide really including and how it is made of, as well as the capture kinetics.
Meanwhile, I think they should consider to cite a very recent publication from IJMS, a MDPI journal, to recognzie the similar effort made by other research groups (Jou HJ et al, IJMS 2022, pubmed id 36499466).
Author Response
Dear Reviewer,
Thank you very much for your time, positive and stimulating comments.
- We have improved our work with a new paragraph in Materials and Methods, in which we have described the SBS Slide technology and its validation (2.3 SBS ® Slide Technology text lines 98-138).
- We have cited the new very interesting paper by Jou HJ et al, IJMS 2022 (text lines 403-405, reference n. 27).
The revisions to our manuscript have been marked up using the “Track
Changes” function of MS Word.
We have added a new figure, figure 1, so all other figures have been updated up to number 7.

Round 2
Reviewer 1 Report
The authors addressed well the main issues.